# Encapsulated Essential Oils Improve the Growth Performance of Meat Ducks by Enhancing Intestinal Morphology, Barrier Function, Antioxidant Capacity and the Cecal Microbiota

**DOI:** 10.3390/antiox12020253

**Published:** 2023-01-22

**Authors:** Hongduo Bao, Yongqiang Xue, Yingying Zhang, Feng Tu, Ran Wang, Yu Cao, Yong Lin

**Affiliations:** 1Institute of Food Safety and Nutrition, Key Lab of Food Quality and Safety of Jiangsu Province-State Key Laboratory Breeding Base, Jiangsu Academy of Agricultural Sciences, Nanjing 210014, China; 2CALID Biotechnology (Wuhan) Co., Ltd., Wuhan 430073, China; 3Institute of Animal Sciences, Jiangsu Academy of Agricultural Sciences, Nanjing 210014, China; 4Institute of Agricultural Facilities and Equipment, The Agriculture Ministry Key Laboratory of Agricultural Engineering in the Middle and Lower Reaches of Yangtze River, Jiangsu Academy of Agricultural Sciences, Nanjing 210014, China

**Keywords:** encapsulated essential oils, growth performance, intestinal morphology, barrier function, antioxidant capacity, cecal microbiota

## Abstract

The objective of this study was to evaluate the effects of encapsulated essential oils (EOs) on the gut microbiota, growth performance, intestinal morphology, antioxidant properties and barrier function of meat-type ducks. A total of 320 male Cherry Valley ducks (1 day old), were randomly assigned to four dietary experimental groups with eight replicates of ten ducks each. The groups consisted of the CON group (basal diet), the HEO group (basal diet + EO 1000 mg/kg), the LEO group (basal diet + EO 500 mg/kg), and the ANT group (basal diet + chlortetracycline 50 mg/kg). Our findings indicated that ducks fed with EO 1000 mg/kg had greater average daily feed intake (ADFI), average daily gain (ADG), and body weight (BW) and a lower feed conversion ratio (FCR) than the other groups. The serum concentration of TG reduced in the HEO (*p* > 0.05) and LEO (*p* < 0.05) groups on day 42, while the concentration of CHOL increased with the EO concentration in the LEO (*p* > 0.05) and HEO (*p* < 0.05) groups. No differences were observed in the ileal mucosa for the activities of SOD, MPO and GSH-PX after EO dietary treatment. Dietary supplementation with EOs significantly increased the villus heights (*p* < 0.01) and the ratio of villus height to crypt depth (*c/v*) in the duodenum and jejunum of ducks. Moreover, the mRNA expressions of *Claudin*1 and *Occludin* in the jejunal mucosa were observed to be higher in the LEO and HEO groups rather than the CON and ANT groups on d 42. The α diversity showed that the HEO group improved the bacterial diversity and abundance. The β diversity analysis indicated that the microbial structures of the four groups were obviously separated. EO dietary supplementation could increase the relative abundance (*p* < 0.01) of the *Bacteroidetes* phylum, *Bacteroidaceae* family, and *Bacteroides*, *Desulfovibrio*, *Phascolarctobacterium*, and *Butyricimonas* genera in the cecal microbiota of ducks. We demonstrated significant differences in the bacterial composition and functional potential of the gut microbiota in ducks that were fed either an EO diet or a basal diet. Therefore, supplemented EOs was found to have a positive effect on the growth performance and intestinal health of ducks, which was attributed to the improvement in cecal microbiota, intestinal morphology, and barrier function.

## 1. Introduction

With the deterioration of the unsurpassed negative side of antibiotic and the development of multi-resistant pathogenic bacteria, essential oils (EOs) are being researched as a possible substitute for antibiotics [1,2]. EOs, composed of volatile compounds such as terpenes and their derivatives, are now a sought-after feed additive for broiler chickens, pigs [3], beef cattle [4] and sheep [5]. EOs are derived from folk medicine, making them environmentally friendly, non-toxic and in harmony with nature, and they are generally regarded as safe (GRAS) for mammals [6]. Previous bioassays affirmed that EOs possess significant antibacterial, antioxidant, and antifungal activities [7,8,9,10,11,12]. Moreover, it has been demonstrated that EOs can stimulate the secretion of digestive enzymes, which can then facilitate the digestion and absorption of nutrients, leading to improved animal production performance [13]. Therefore, in recent years, the utilization of EOs as feed additives has been widely acknowledged as a viable alternative to antibiotic growth promoters (AGPs), which have been banned in the European Union since 2006 and in China in 2020 [14,15].

The growth performance of ducks is heavily reliant on gut health, which promotes nutrient absorption and offers protection from disease [16]. Any factor influencing gut health will surely have an effect on the animal as a whole, thus inevitably changing its nutrient uptake and needs [17]. The balance of the gut microbiota is an important factor for maintaining gut health in animals, as it optimizes digestion and absorption, regulates energy metabolism, prevents mucosal infections, and modulates the immune system, thereby keeping the host in a state of homeostasis [18]. Previous research has indicated that thymol and cinnamaldehyde, possessing antibacterial properties, can enhance nutrient digestibility and production parameters in pigs and broilers, along with modifying intestinal morphology and the gut microflora barrier. Nonetheless, there is still a scarcity of in vivo experiments conducted in ducks [19,20]. Moreover, EOs are volatile and thermally unstable and may lose their functional activities during pelleting and feeding processing, as well as through improper storage [21]. Consequently, to prevent volatilization and preserve the active ingredients of EOs, encapsulation is the most widely employed technique to mask undesirable smells and enhance solubility [22]. There is evidence that the effects of dietary EOs on pig and poultry’s gut microbiota may be different, or even contradictory [23,24,25]. Furthermore, one comprehensive review article elucidated the inconsistent data regarding the intricate gut environment and the mechanisms involved in the growth improvement between species and within the same species [14]. Considering these problems, an essential oil coated with a safe food emulsifier (glycerol monolaurate, a monoglyceride composed of lauric acid and glycerol that has been approved by the US Food and Drug Administration) could be an ideal feed additive for meat ducks. Therefore, our objective here was to evaluate the influence of encapsulated EOs on the growth performance and gut health of meat-type ducks, with a particular emphasis on the alteration of the hindgut microbiota.

## 2. Materials and Methods

### 2.1. Ducks Husbandry and Experimental Design

A total of 320 male Cherry Valley ducks (1 day old) were obtained from a commercial hatchery and housed in experimental facilities with a controlled-temperature room. The ducks were randomly distributed to 4 treatments with 8 replicates, each containing 10 birds. Each replicate was housed in a 2 m × 2 m pen equipped with two water channels and one tubular feeder. Food and water were provided ad libitum throughout the 42 days of the experimental period. The ingredients and calculated nutritive value of the basal diet are presented in Table 1. The nutrient values of the basal diets met the nutrients requirements of ducks as required by Nutrient Requirements of Poultry (NRC) [26]. This experiment was conducted in two stages: the early stage (1–21 days) and the middle–last stage (22–42 days). The experimental treatments were composed of the CON group (basal diet), ANT group (basal diet + 50 mg/kg chlortetracycline), HEO group (basal diet + EO 1000 mg/kg) and LEO group (basal diet + EO 500 mg/kg). The essential oils (product name: Anti-Clos) in the study were supplied by CALID Biotechnology Co., LTD (Wuhan, China). The main active components of the essential oils coated with glycerol monolaurate are glycerol monolaurate (800 g/kg), cinnamaldehyde (54 g/kg) and thymol (6 g/kg). All ducks were treated humanely according to the animal care and ethics committee of the Jiangsu Academy of Agricultural Sciences (SYXK2020-0023), and all efforts were made to minimize animal suffering.

### 2.2. Growth Performance

Individual duck body weight (BW) and pen feed intake were determined every week for the calculation of average daily gain (ADG), average daily feed intake (ADFI), and feed conversion ratio (FCR).

### 2.3. Sample Collection

Except for production performance data, all samples were obtained at 42 days of age of meat ducks. At 42 days of age, one duck per replicate with a body weight close to the mean body weight (8 birds per treatment) was selected. Blood samples were collected by puncturing the veins of wings and clotting in polypropylene tubes containing heparin sodium. Serum samples were separated after being centrifuged at 3000× r/min, kept at 4 °C for 15 min and frozen at −20 °C for subsequent analysis. Afterwards, these ducks were euthanized with bleeding of the carotid artery. Subsequently, approximately 1 cm segments from the median sections of the duodenum and jejunum were collected and preserved in 4% paraformaldehyde solution for intestinal morphological measurements. The mucosa of the jejunum and ileum was scraped from the middle portion and stored in sterile centrifuge tubes and snap-frozen in liquid nitrogen at −80 °C for the assay of antioxidant indexes and barrier function. The cecum content was aseptically collected into sterile tubes and frozen at −80 °C for evaluating gut microbial composition.

### 2.4. Serum Biochemical Parameters Analysis

The activity of serum glucose (GLU), triglyceride (TG), cholesterol (CHOL), total protein (TP), blood urea nitrogen (BUN), alanine amiotransferase (ALT), alkaline phosphatase (ALP), lactic dehydrogenas (LDH) and aspartate aminotransferase (AST) was determined by an automatic biochemical analyzer (HITACHI 7020). All the above indicators were determined using kits provided by MedicalSystem Biotechnology Co., LTD (Ningbo, China).

### 2.5. Antioxidant Properties of Ileal Mucosa

Ileal mucosa was homogenized in ice-cold phosphate-buffered saline (PBS) and then centrifuged at 10,000× *g* at 4 °C for 10 min, and the supernatant was used to determine the activities of superoxide dismutase (SOD), myeloperoxidase (MPO) and glutathione peroxidase (GSH-Px). These indicators were determined using kits provided by Jiancheng Bioengineering Institute (Nanjing, China) according to the manufacturer’s instructions.

### 2.6. RNA isolation and Quantitative Real-Time PCR

Total RNA was extracted from the jejunal mucosa using an RNA isolation kit (Beijing Transgene Biotech Ltd., Beijing, China) following the manufacturer’s protocol. Then, 1 μg of total RNA from each sample was used to generate cDNA in a final volume of 20 μL using a PrimeScriptRT Reagent Kit with a gDNA Eraser (Beijing Transgene Biotech Ltd., Beijing, China). Quantitative real-time PCR was performed using SYBR Green supermix (Beijing Transgene Biotech Ltd., Beijing, China) on a LightCycler 480 (Roche, Basel, Switzerland), and primers targeting mucosal barrier functions are shown in Table 2. Each experiment was performed in triplicate. The β-actin gene was used as an internal standard, and the relative mRNA expression levels were calculated by the 2-ΔΔCt method [27].

### 2.7. Morphological Analysis of Duodenum and Jejunum

The duodenal and jejunal samples were dehydrated and embedded in paraffin wax. Paraffin sections of tissues were sectioned at 5 μm thickness, stained with hematoxylin and esosin, and observed with a microscope. Representative fields were selected and photographed. The average intestinal villus height (VH) and crypt depth (CD) were measured using Image J software (v1.8.0, National Institutes of Health, Betheseda, MD, USA), and the villus height/crypt depth (V/C) was calculated.

### 2.8. 16S rRNA Gene Sequencing and Analysis

The total genomic DNA of 24 cecal content samples was extracted using TianmoTM DNA Isolation Kit of Stool (Beijing Tianmo Sci & Tech Development Co., Ltd., Beijing, China). The extracted DNA was measured using a NanoDrop ND-1000 spectrophotometer (Thermo Fisher Scientific, Waltham, MA, USA) and agarose gel electrophoresis, respectively. The V3–V4 regions of bacteria 16S ribosomal RNA were amplified using polymerase chain reaction (PCR) with primers with the barcode (F: CCTACGGGNGGCWGCAG, R: GACTACHVGGGTATCTAATCC). The purified amplicon samples were analyzed by Tianhao Biotechnology Co. Ltd. (Shanghai, China) using high-throughput sequencing (2 × 250 bp, paired-end sequence) on the Illumina NovaSeq 6000 (Illumina, San Diego, CA, USA).

The data were arranged by changing files, demultiplexing and qualifying the acquired Illumina reads using QIIME (version 1.17) [28]. Operational taxonomic units (OTUs) were aggregated at a similarity level of 0.97 using UPARSE. The α diversity indexes including observed, Chao1, ACE, Shannon, Simpson and Good’s coverage were analyzed. Principal coordinate analysis (PCoA) and PLS-DA of the microbial community relationship were performed with STAMP software using Jaccard as the statistical test.

### 2.9. Statistical Analysis

The data of the growth performance, duodenal and jejunal morphology, serum biochemical indexes, antioxidant index, and mRNA expression were analyzed using one-way analysis of variance (ANOVA) with Duncan’s multiple range tests using SPSS Statistics 16.0 (SPSS Inc., Chicago, IL, USA). Data were shown as mean values with standard error. For all tests, a probability level of *p* < 0.05 was considered as a statistically significant difference, while *p <* 0.01 was a very significant difference.

## 3. Results

### 3.1. Growth Performance

The effects of EO treatments on growth performance of ducks are shown in Table 3. Overall, the EO treatments of both concentrations greatly affected the production performance of the meat ducks, especially in the early stage (1–21 days). Compared with the basal diet and antibiotic diet, 1000 mg/kg of EO (HEO group) increased the ADFI, ADG, and BW during 1–42 d (*p* < 0.05). Moreover, FCR improved significantly (*p* < 0.05) in the early stage (1–21 days) and overall (1–42 days) in the HEO group compared to the CON group. However, dietary supplementation with 500 mg/kg of EO (group LEO) had no significant effect on the FCR of meat ducks during the two stages (*p* > 0.05). Furthermore, 500 mg/kg of EO (LEO group) only significantly enhanced the ADG during 1–21 d (*p* < 0.05) and 1–42 d (*p* < 0.05), and BW of meat ducks during 1–42 d (*p* < 0.05). 

### 3.2. Serum Biochemical Indexes

The serum biochemical indexes of meat-type ducks are shown in Table 4. The activities of GLU, TP, BUN, ALT, and LDH were not markedly affected by dietary EO (*p* > 0.05). The serum concentration of AST reduced in the LEO (*p* > 0.05) and HEO (*p* < 0.05) groups on day 42 compared with the CON group. Moreover, compared to the CON group, the concentration of TG reduced in the HEO (*p* > 0.05) and LEO (*p* < 0.05) groups. However, the concentration of CHOL increased with the EO concentration in the LEO (*p* > 0.05) and HEO (*p* < 0.05) groups.

### 3.3. Antioxidation of Ileal Mucosa

As shown in Figure 1, no treatment differences were detected in the activity of SOD, MPO, and GSH-PX on day 42 in the ileal mucosa (*p* > 0.05). However, the activity of MPO was enhanced in both the EO1000 and EO500 groups (*p* > 0.05), and the higher concentration of EO was associated with higher MPO activity.

### 3.4. Duodenal and Jejunal Morphology

Compared with the CON group, dietary supplementation of EO increased the villus heights (*p* < 0.01) and the ratio of villus height to crypt depth (*c/v*) (*p* < 0.01) in the duodenum and jejunum (Figure 2), but had no significant effects on the crypt depth in the duodenum and jejunum (*p* > 0.05), while dietary supplementation of chlortetracycline did not affect the duodenal and jejunal morphology (*p* > 0.05).

### 3.5. Effects of EO on the Gene Expression of Barrier Functional Proteins

The gene expressions of intestinal barrier function proteins are shown in Figure 3. Compared with the basal diet group (CON group), adding different levels of EO increased the mRNA expression of Claudin1 and Occludin in the jejunal mucosa. When the added amount of EO reached 500 mg/kg (LEO group), the gene level of Claudin1 significantly improved (*p* < 0.01). Supplementation with 1000 mg/kg of EO (HEO group) significantly enhanced the gene expression of Occludin (*p* < 0.05). In addition, there were no treatment differences in the gene expression of ZO-1 (*p* > 0.05).

### 3.6. Cecal Microbiota Analysis

The OTUs were classified into 6728 amplicon sequence variants (ASVs). The rarefaction curves (Appendix A) tended towards a saturated plateau. This shows that the microbial communities of the 24 samples were deep enough to estimate phenotypic richness and microbial community diversity.

### 3.7. Variation in Alpha and Beta Diversity

The coverage indices were 0.999 or 0.998 in the four groups, indicating that the sequencing depth of each sample was sufficient. The observed, Chao1, ACE, and Shannon indexes all increased, and the Simpson index decreased in the HEO and LEO groups compared to the CON and ANT groups; moreover, the Shannon index increased significantly in the HEO group (*p* < 0.05) (Figure 4A). α diversity shows that the HEO group (EO1000) has improved diversity and abundance of the gut microflora. To analyze the inter-individual differences, principal co-ordinates analysis (PCoA) and partial least squares discriminant analysis (PSL-DA) (Figure 4C) were used. The PCoA based on the Jaccard index showed distinct separations between the relative abundance of microbial community from the four groups, but the microbial composition was similar within groups. Moreover, PLS-DA further determined the differences in the compositions of gut microbiota between the four groups.

### 3.8. Common and Unique Microbial Populations

The microbial similarity between treatments based on the common ASVs are shown in the Venn diagram (Figure 4D). A total of 336 ASVs were common to all groups. After improving the EO concentration, the number of ASVs present only in one group increased from 660 to 861.

A complete list of the sequences identified (relative abundance) at the phylum levels is provided in Table 5, and the representative phyla with significant differences between groups are shown in Figure 4B. Firmicutes was the most abundant phylum in the CON (55.43%) and ANT (51.10%) groups. However, ducks that received EOs showed reductions in this phylum (23.22% of the HEO group and 27.33% of the LEO group), with more *Bacteroidetes*, *Proteobacteria*, *Fusobacteria*, and *Deferribacteres* and less *Actinobacteria*. At the family level, a total of 105 families were identified (Figure 4E and Appendix A). The ducks that received EOs showed lower percentages of *Verrucomicrobiaceae* (*p* > 0.05), *Ruminococcaceae* (*p* > 0.05) and *Peptostreptococcaceae* (*p* < 0.01). The abundance of the family *Bacteroidaceae* was significantly increased in the HEO and LEO groups compared to the CON and ANT groups (*p* < 0.01). Differences in the four groups referring to the representation of the 11 genera are shown in Figure 4F. Compared to the CON group, the abundance of the families *Bacteroides*, *Desulfovibrio*, *Phascolarctobacterium*, *Acetanaerobacterium*, *Butyricimonas*, *Anaerobiospirillum*, *Alistipes*, and *Ruminococcus2* were increased in the HEO and LEO groups (*p* < 0.05 or *p* < 0.01 or *p* < 0.001). Conversely, *Romboutsia*, *Akkermansia*, and S*ubdoligranulum* were visibly reduced when EOs were used as feed additives for ducks.

Using the picrust2 analysis tool, functions of cecal microbiota were predicted and analyzed based on the 16SrRNA sequencing data. As shown in Figure 4G, the functional abundances of the Kyoto Encyclopedia of Genes and Genomes (KEGG)_pathway of all samples are displayed in a heatmap. A total of 127 pathways were predicted according to all samples, with 108 pathways enhanced after EO treatment (*p* < 0.05 or *p* < 0.01) (Appendix A). In this study, ko00020: citrate cycle (TCA cycle); ko00740: riboflavin metabolism; ko00511: other glycan degradation; and ko00720: carbon fixation pathways in prokaryotes were predominant across all samples. The significant differences in the citrate cycle, lipoic acid metabolism, phenylalanine, tyrosine and tryptophan biosynthesis, DNA replication and protein degradation and adsorption in the EO groups compared to the CON group are shown in Figure 4H.

## 4. Discussion

EOs have been broadly applied as antibacterial and antioxidant feed additives in pigs and poultry, yet their application in meat-type Cherry Valley ducks has been less explored. Our experiment here sought to evaluate the practicality of dietary supplementation with 500 mg/kg or 1000 mg/kg of an encapsulated blend of EOs in terms of performance, gut health, and gut microbial assessment in comparison to an antibiotic diet and a baseline diet. In the current study, compared with the CON and ANT groups, significant differences (*p <* 0.05) occurred in ADG, BW and FCR with supplementation of 1000 mg/kg of EOs, but 500 mg/kg of EOs did not significantly decrease the FCR of meat ducks during the two phases (*p* > 0.05). Previous research also indicated that adding 600 mg/kg of EOs significantly increased the ADG and BW and lowered the FCR during the whole growing period of AA broilers [29]. In line with our study, Mohiti-Asli et al. (2018) and Mohiti-Asli et al. (2017) showed that supplementation of oregano EOs at 300 mg/kg in the diet of broilers had beneficial effects on their performance and immune system [30,31]; Ruan et al. (2021) found that the weight, ADFI and ADG of yellow broilers at 30 days of age were significantly augmented when 150 and 300 mg/kg oregano EOs were added [32]. Youssef et al. (2020) determined that adding a 25 mg/kg mixture of star anise, rosemary and thyme oregano EOs could significantly enhance the body-weight gain of broilers [33]. In contrast to these studies, Park et al. (2015) found that the growth performance of meat ducks did not show significant differences when different doses of oregano dry powder were added to the meat duck diet [34]. Hashemipour et al. (2013) reported that addition of carvacrol and thymol in animal feed decreased the ADFI [35]; Abouelezz et al. (2019) found that there was no significant improvement in the growth performance of Cherry Valley meat ducks when fed a diet supplemented with 50 mg/kg and 100 mg/kg of oregano EOs, or 150 and 300 mg/kg of EOs [36]. Xue et al. (2020) also found that a diet supplemented with a 100 mg/kg EO mixture containing eucalyptus oil, carvacrol, cinnamaldehyde and capsaicin had no significant effect on the growth performance, carcass traits and intestinal morphology of broilers [37]. The large discrepancies between the aforementioned studies could be attributed to a variety of factors, such as the type of essential oils used, the coating technology employed, the composition of the ingredients, the dosage administered, and the variety and age of the experimental animals.

It is well known that an increase in the villi height and the ratio of villi height to crypt depth, alongside a decrease in crypt depth, will lead to a larger mucosal surface area in the small intestine, thereby increasing its digestive capacity. In our experiment, ducks that were fed 500 or 1000 mg/kg of EOs in their diet had a higher VH and V/C ratio in the duodenum and jejunum than those that were fed either the basal diet or the antibiotic diet, which can partially explain the improvements in ADG, BW and FCR. Our results corroborated Ding et al.’s (2022) discovery that the jejunal morphology was enhanced when dietary EO supplement was given at 200, 400 and 600 mg/kg.

The intercellular tight junctions are the major determinants of the intestinal physical barrier, composed of junctional molecules such as claudin, occludin and zonula occludins [38]. A beneficial effect on the expression of tight junction proteins in the jejunum (Claudin-1, ZO-1, mucin-1) and ileum (Occludin, Claudin-1, mucin-2) was observed in piglets fed microencapsulation EOs (1500 mg/kg) [39]. Liu et al. (2018) found that the gene expressions of Occludin, Claudin-1, Claudin-5, ZO-1, and ZO-2 in the intestinal mucosa of Ross 308 broilers were significantly increased when they were administered diets containing 200, 300, and 400 mL of carvonol EOs orally for two weeks [40]. Our study demonstrated that claudin-1 or occludin levels were upregulated in meat ducks that were supplied with EO. This suggests that EOs can directly promote intestinal health by reinforcing the intestinal barrier function.

The cecum harbors an intricate, diverse, and sustained community of microbiota. The cecal microbiota is integral to the health of the animal gut as it can enhance absorption efficiency, counteract the effects of pathogenic bacteria, bolster intestinal integrity, and modulate immunity. Previous studies have indicated that EOs positively affected the structure of the intestinal microflora of livestock, such as by modulating the diversity and composition of the gut microbiota, improving the relative abundance of beneficial bacteria and reducing the proportion of harmful bacteria [41,42]. For example, the gut microbiota of broilers has been altered by adding plant extracts containing thymol and carvacrol into the diet. In pigs, EOs also enhanced the abundances of the *Actinobacteria* phylum and *Bifidobacterium* genera and lowered (*p* < 0.05) the *Bacteroidetes* phylum and *Alloprevotella* genera in the colonic digesta [29]. In our studies, there were significant differences in the abundance and diversity in the HEO group compared to the CON group. Moreover, PCoA and PLS-DA were used to distinguish and cluster the microbial community of each group, implying that encapsulated EOs could modify the gut microbiota of ducks. The α diversity of cecal microbes in the EO dietaries was higher, which concurred with Ding et al., as observed in this study [42]. Β diversity of the microbiota varied between treatments, which corroborates the results of Ma et al. In animal gut, the phyla *Bacteroidetes* and *Firmicutes* were dominant. Our results also revealed that *Firmicutes* was the dominant phylum supplied with the basal diet of the ducks, and the abundance of *Bacteroidetes* and *Proteobacteria* was significantly higher after the EO dietary treatment. Our results were consistent with earlier studies that investigated *Bacteroidetes* as the dominant phylum and *Bacteroides* as the dominant genus in the cecum [43]. The *Bacteroidetes* phylum is a prevalent and greatly varied phylum in mammal intestines. The members of this phylum have the ability to decompose complex components, i.e., carbohydrates and proteins, allowing their hosts to derive nourishment from their diets [44]. Additionally, they exhibit a significant ability to respond to the stress imposed by the gut and host environment, and its abundance is correlated with the amount of SCFA content [45]. The differences could be attributed to the breed, age, or environment.

Our study found that EO exposure changed the microbial composition and the potential function of the cecal microbiota of meat ducks, resulting in a different intestinal environment. These findings may account for the improved growth performance observed in the growing period. The results obtained will be utilized to assess the efficacy of EOs as a feed additive in terms of animal welfare, performance, and economic benefits.

## 5. Conclusions

In conclusion, the present study demonstrated that dietary supplementation of encapsulated EOs at 500mg/kg and 1000 mg/kg improved growth performance, favorably affected gut morphology, and enhanced the gene expression of intestinal barrier function proteins in meat-type ducks. It is noteworthy that dietary EOs at 1000 mg/kg could significantly decrease the FCR of ducks. Furthermore, feeding the ducks with the EO diet significantly changed the composition of bacterial flora and enhanced the potential function of the cecal microbiota. Overall, supplementing duck diets with encapsulated EOs could be a beneficial approach to ensure optimal growth and gut health, both in terms of gut morphology and gut microbiology.

## Figures and Tables

**Figure 1 antioxidants-12-00253-f001:**
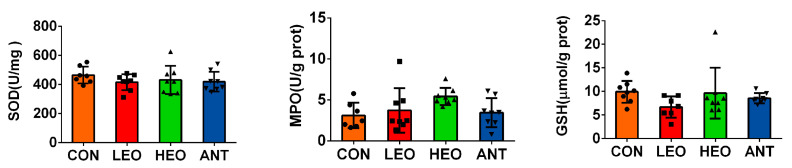
Antioxidant activities of ileal mucosa. Note: CON, meat ducks fed a basal diet; LEO, meat ducks fed a basal diet supplemented with 500 mg/kg EO coated with glycerol monolaurate; HEO, meat ducks fed a basal diet supplemented with 1000 mg/kg EO coated with glycerol monolaurate; ANT, meat ducks fed a basal diet supplemented with 50 mg/kg chlortetracycline. SOD, superoxide dismutase; MPO, myeloperoxidase; GSH, glutathione peroxidase.

**Figure 2 antioxidants-12-00253-f002:**
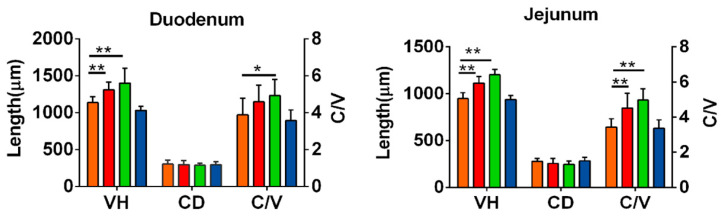
Duodenal and jejunal morphology. Note: VH, villus height; CD, crypt depth; *c/v*: the ratio of villus height to crypt depth. Orange, group CON; red, group LEO; green, group HEO; blue: group ANT. “*”, *p* < 0.05; “**”, *p* < 0.01.

**Figure 3 antioxidants-12-00253-f003:**
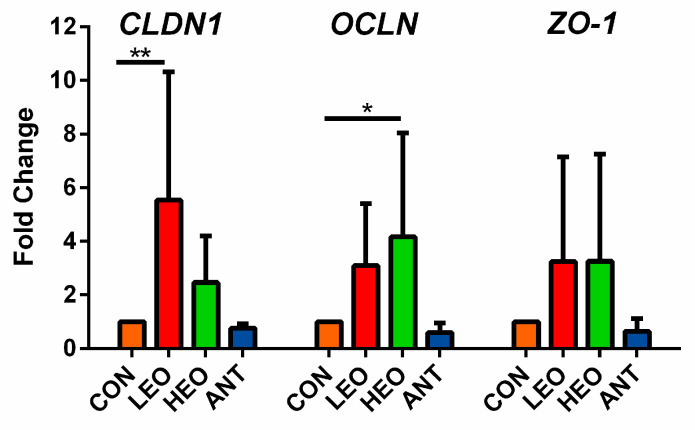
Relative mRNA expressions of *Claudin-1*, *Occludin*, and *ZO-1* in jejunal mucosa. “*”, *p* < 0.05; “**”, *p* < 0.01.

**Figure 4 antioxidants-12-00253-f004:**
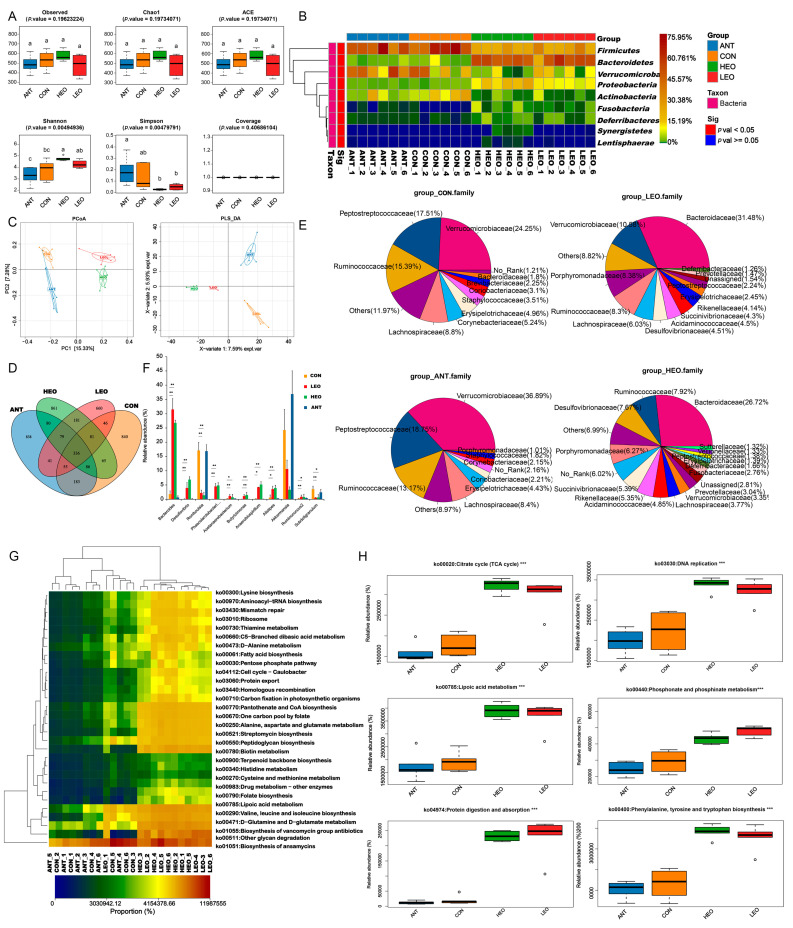
Analysis of cecal microbiota. (**A**) α diversity analysis using observed, Chao1, ACE, Shannon, Simpson, and coverage indexes between the four groups. For ^a, b, c^ above the column, different letters mean significant differences (*p* < 0.05). (**B**) Representative phyla based on OTUs identified in samples from each duck. (**C**) β diversity analysis. PCoA and PLS-DA plots were all based on the abundances of OTUs, and the percentage in x- and y-axes means contribution value to the discrepancies by the component. Dots represent samples. Samples in the same group share the same color. (**D**) Each circle in the Venn diagram represents one group noted by the name of same color. The numbers located in the overlapping area represent the number of OTUs shared with respective groups. The numbers located in the individual areas represent the number of OTUs separated to the representative group. (**E**) The relative abundance of family-level diversity of each group. Shown are representative families based on OTUs identified in samples from each duck; the average OTU coverage of each group is displayed as a pie plot. (**F**) Differences of the three experimental groups (LEO, HEO and ANT) compared to the CON group, which referred to the representative genus. **, *p* < 0.001; *, *p* < 0.05. (**G**) The functional abundances of KEGG_pathway of all samples are displayed in a heatmap. (**H**) Six representative different functional abundances of the identified KEGG_pathways (***, *p* < 0.01) are shown here.

**Table 1 antioxidants-12-00253-t001:** Composition and nutritional information of the basal experimental diets.

Items	Day 1–21	Day 22–42
**Ingredients (%)**		
Corn	45.83	46.00
Rice barn	7.50	10.00
Wheat meal	12.00	10.00
Wheat bran	3.00	5.00
Soybean meal	15.70	6.20
Corn gluten meal	3.00	4.50
Peanut meal	2.50	2.50
Soybean oil	0.30	3.00
DDGS	5.50	8.00
CaHPO_4_	0.72	0.75
Limestone	1.40	1.40
Lysine	0.05	0.15
Premix ^1^	2.50	2.50
Total	100.00	100.00
**Calculated Values**		
Gross energy (MJ/kg)	11.8	12.5
Crude protein (%)	19.50	18.50
Calcium (%)	0.85	0.75
Available phosphorus (%)	0.38	0.33
Lysine	1.20	1.15
Methionine (%)	0.60	0.40
Methionine + Cystine (%)	0.95	0.71

^1^ Provided per 1 kg of diet: vitamin A, 10,000 IU; vitamin D_3_, 3750 IU; vitamin E, 16.25 mg; vitamin K3, 2.0 mg; vitamin B_1_, 3.5 mg; vitamin B_2_, 7.5 mg; vitamin B_6_, 3.75 mg; vitamin B_12_, 0.02 mg; folic acid, 1.0 mg; biotin, 0.13 mg; niacin, 40.0 mg; pantothenic acid, 11.25 mg; copper, 6 mg; iron, 40 mg; zinc, 65 mg; manganese, 80 mg; iodine, 0.5 mg; selenium, 0.3 mg.

**Table 2 antioxidants-12-00253-t002:** Primers used in this study.

Primer Name	Sequence (5′–3′)	Amlicon Size (bp)	Gene ID
β-actin	CAGCCAGCCATGGATGATGA	137	NM_205518.2
	ACCAACCATCACACCCTGAT		
*CLDN1*	CATACTCCTGGGTCTGGTTGGT	100	CP100563.1
	GACAGCCATCCGCATCTTCT		
*OCLN*	ACGGCAGCACCTACCTCAA	945	CP100582.1
	GGGCGAAGAAGCAGATGAG		
*ZO-1*	CCACTGCCTACACCACCATCTC	138	CP100564.1
	CGTGTCACTGGGGTCCTTCAT		

**Table 3 antioxidants-12-00253-t003:** Effects of dietary EO supplementation on growth performance of ducks.

Items ^1^	CON	ANT	HEO	LEO
D 1–21				
ADFI, g/d	79.43 ± 4.04	83.22 ± 1.13	85.93 ± 2.29	83.60 ± 3.38
ADG, g/d	51.39 ± 1.41 ^c^	55.60 ± 0.66 ^b^	60.31 ± 1.43 ^a^	56.27 ± 0.68 ^b^
FCR	1.53 ± 0.01 ^a^	1.50 ± 0.01 ^a^	1.43 ± 0.02 ^b^	1.49 ± 0.01 ^a^
D 22–42				
ADFI, g/d	268.51 ± 2.36	268.57 ± 2.66	277.28 ± 4.39	267.54 ± 1.44
ADG, g/d	144.21 ± 0.49	145.71 ± 0.73	148.09 ± 1.12	144.50 ± 0.97
FCR	1.90 ± 0.03	1.87 ± 0.02	1.87 ± 0.03	1.89 ± 0.02
D 1–42				
ADFI, g/d	171.66 ± 1.15 ^b^	173.64 ± 1.30 ^b^	179.12 ± 1.76 ^a^	174.30 ± 0.93 ^b^
ADG, g/d	96.67 ± 0.56 ^c^	99.56 ± 0.60 ^b^	102.27 ± 0.39 ^a^	99.31 ± 0.42 ^b^
BW, kg	4.01 ± 0.02 ^c^	4.13 ± 0.02 ^b^	4.24 ± 0.02 ^a^	4.12 ± 0.02 ^b^
FCR	1.82 ± 0.02 ^a^	1.77 ± 0.01 ^b^	1.76 ± 0.01 ^b^	1.79 ± 0.02 ^ab^

Note: ^1^ ADFI, average daily feed intake; ADG, average daily gain; BW, body weight; mean ± S.D. values are expressed in the table; CON, meat ducks fed a basal diet; ANT, meat ducks fed a basal diet supplemented with 50 mg/kg chlortetracycline; HEO, meat ducks fed a basal diet supplemented with 1000 mg/kg EO coated with glycerol monolaurate; LEO, meat ducks fed a basal diet supplemented with 500 mg/kg EO coated with glycerol monolaurate. For ^a,b,c^ within rows, different letters indicate significant differences (*p* < 0.05), and having the same letter means the difference is not significant (*p* > 0.05).

**Table 4 antioxidants-12-00253-t004:** Effects of EO on serum biochemical parameters of ducks.

Items	CON	ANT	HEO	LEO
GLU	8.40 ± 0.27	8.71 ± 0.27	8.85 ± 0.07	9.05 ± 0.25
TG	0.84 ± 0.06 ^b^	0.79 ± 0.06 ^b^	0.72 ± 0.04 ^ab^	0.60 ± 0.04 ^a^
CHOL	5.52 ± 0.23 ^a^	5.63 ± 0.13 ^ab^	6.07 ± 0.17 ^b^	6.00 ± 0.10 ^ab^
TP	47.73 ± 1.14	49.61± 0.97	47.88 ± 0.8	47.91 ± 0.66
BUN	0.64 ± 0.03	0.66 ± 0.06	0.60 ± 0.05	0.71 ± 0.09
ALT	54.29 ± 3.26	52.13 ± 4.25	49.88 ± 1.91	51.14 ± 2.58
AST	19.29 ± 1.43 ^b^	18.88 ± 2.82 ^ab^	13.88 ± 1.27 ^a^	17.43 ± 0.78 ^ab^
ALP	33.71 ± 4.82 ^ab^	28.13 ± 2.67 ^a^	33.63 ± 1.81 ^ab^	38.71 ± 4.73 ^b^
LDH	411.71 ± 36.19	418.00 ± 48.84	370.00 ± 21.50	403.43 ± 26.69

Note: GLU, glucose; TG, triglyceride; CHOL, cholesterol; TP, total protein; BUN, blood urea nitrogen; ALT, alanine aminotransferase; AST, aspartate aminotransferase; ALP, alkaline phosphatase; LDH, lactic dehydrogenase. For ^a,b^ within rows, different letters indicate significant differences (*p* < 0.05), and having the same letter means the difference is not significant (*p* > 0.05).

**Table 5 antioxidants-12-00253-t005:** The cecal microbiota composition at phylum level (%).

Phylum	ANT	CON	HEO	LEO
*Firmicutes*	51.10 ± 13.70	55.43 ± 14.33	23.22 ± 3.05 **	27.33 ± 7.36 *
*Bacteroidetes*	2.46 ± 1.72	3.53 ± 3.62	49.41 ± 3.47 **	47.23 ± 13.86 **
*Verrucomicrobia*	36.90 ± 16.24	24.25 ± 20.23	3.34 ± 4.52	10.61 ± 7.73
*Proteobacteria*	1.59 ± 0.44	2.38 ± 0.72	16.97 ± 4.75 **	10.70 ± 1.73 **
*Actinobacteria*	7.45 ± 8.19	13.86 ± 7.54	1.38 ± 1.56 *	1.90 ± 3.11 *
*Fusobacteria*	0.04 ± 0.05	0.006 ± 0.01	2.77 ± 3.21 *	0.81 ± 0.63
*Deferribacteres*	0.11 ± 0.15	0.13 ± 0.14	1.67 ± 1.63	1.26 ± 1.17
*Unassigned*	0.24 ± 0.10	0.34 ± 0.19	0.12 ± 0.35	0.09 ± 0.05

Note: The LEO, HEO and ANT groups were compared to the CON group with statistical significance set at *p* < 0.01 (**) and *p* < 0.05 (*).

## Data Availability

The sequencing data from this study have been submitted to the NCBI Sequence Read Archive (http://www.ncbi.nlm.nih.gov/bioproject/899684) under accession PRJNA899684 on 9 November 2022.

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
