# Peer review of "Encapsulated Essential Oils Improve the Growth Performance of Meat Ducks by Enhancing Intestinal Morphology, Barrier Function, Antioxidant Capacity and the Cecal Microbiota"

_antioxidants, 2023, doi:10.3390/antiox12020253_

Round 1

Reviewer 1 Report

This manuscript investigates the growth performance of meat ducks with the effects of encapsulated essential oils on gut morphology, barrier function, antioxidant capacity, and cecal microbiota. It is an attractive, innovative application with academic value research. This manuscript is suitable for publication after modification.

1. Line 95-98. The description of the sentence needs to be clarified. The ingredient description of the encapsulated essential oil (product name: Anti-Clos) cannot be seen. Is the essential oil component a single or multiple compounds? Why choose this product? In short, sufficient and necessary explanations should be made for the important essential oil formula components before they can be discussed. For example, it can be known from Table 3 that JYG improves appetite but does not increase FCR capacity. But why? In other words, without explaining the ingredients of the added essential oils, how to further clarify the research literature and in-depth discussions on the product’s anti-oxidation, protection of epithelial tissue, and regulation of intestinal bacteria?

2. The research method needs to indicate whether the JC and KSS group feeds have the formula of adding coated commercial essential oils. For example, glycerol monolaurate, cinnamaldehyde, thymol, glyceryl dilaurate, glyceryl trilaurate, and inert carriers.

3. Lines 189-192. The statistical labeling in Table 3 needs to be completed; for example, why there are no statistical results for D1-21 and D22-42. In addition, the statistical explanations in the footnotes below the table could be more precise.

4. Line 200-203. The statistical labeling in Table 4 needs to be completed, such as why there are no statistical results for GLC, TP, BUN, ALT, and LDH. In addition, the statistical explanations in the footnotes below the table need to be clarified.

5. Line 208-210. The narrative of the sentence needs to be carefully and precisely corrected.

6. In Figure 2, it seems that the KSS group is not included in the statistical items, and it is suggested that it should be explained where appropriate.

7. Line 225-230. Figure 3 statistical labeling is inaccurate and should be improved.

8. Table 5 needs substantial improvement. The presentation of phylum level (%) classification is messy, irregular, and difficult to understand. The author only explained some of the bacterial phyla and left most of the remaining phyla levels (%) silent Also, why is the statistical analysis only partial? Are the other phylum level (%) data listed in the table not critical?

- Why is Deferribacteres, Candidatus_Saccharibacteria, SR1 listed here?

- The application of statistical methods needs to be clarified. For example, which group is the group with a statistically significant difference (*, **) compared with? Table S1 also has the same problem of unclear application of statistical methods.

9. Line 337-347. This paragraph should not just cite the same phenomenon in other research places but should increase the discussion of the role of commercial EOs on Tight-junction protein (Claudin-1, Occludin, and ZO-1). For example, whether EOs can induce intestinal mucosal antioxidation (Line 348-357) is related and then connect the relationship between intestinal track health and bacterial growth.

Reviewer 2 Report

Dear authors, interesting work as in the subject only a few studies are done. However, some major concerns are present. I suggest to revise carefully the results presentation and specify the EO used. In the attached file my comments

Round 2

Reviewer 2 Report

The paper improved a lot, i suggest for publication